# Intermediate Prototype Mining Transformer for Few-Shot Semantic Segmentation

**Yuanwei Liu**[1][*]    **Nian Liu**[2][*]    **Xiwen Yao**[1]    **Junwei Han**[1][†]

[1]Northwestern Polytechnical University
[2]Mohamed bin Zayed University of Artificial Intelligence
{liuyuanwei9809, liunian228, yaoxiwen517, junweihan2010}@gmail.com

## Abstract

Few-shot semantic segmentation aims to segment the target objects in query under the condition of a few annotated support images. Most previous works strive to mine more effective category information from the support to match with the corresponding objects in query. However, they all ignored the category information gap between query and support images. If the objects in them show large intra-class diversity, forcibly migrating the category information from the support to the query is ineffective. To solve this problem, we are the first to introduce an intermediate prototype for mining both deterministic category information from the support and adaptive category knowledge from the query. Specifically, we design an Intermediate Prototype Mining Transformer (IPMT) to learn the prototype in an iterative way. In each IPMT layer, we propagate the object information in both support and query features to the prototype and then use it to activate the query feature map. By conducting this process iteratively, both the intermediate prototype and the query feature can be progressively improved. At last, the final query feature is used to yield precise segmentation prediction. Extensive experiments on both PASCAL-$5^i$ and COCO-$20^i$ datasets clearly verify the effectiveness of our IPMT and show that it outperforms previous state-of-the-art methods by a large margin. Code is available at https://github.com/LIUYUANWEI98/IPMT

## 1   Introduction

Recent great progress on computer vision rely heavily on a large amount of annotated data, the collecting of which is a time-consuming and labor-intensive work. To solve this problem, few-shot learning is proposed to learn a model that can be generalized to novel categories with only a few annotated images. This setting is also closer to human learning habits which can learn knowledge from scarce annotated examples and identify new categories quickly.

In this paper, we focus on the few-shot semantic segmentation (FSS) task which aims to segment novel objects in the query image with a few annotated support samples. Currently, a lot of works have been proposed for FSS and many of them are based on prototype learning. These methods [39, 28, 12] extract prototypes from the support set to represent the category information and then match them with the query features in a matching network to perform segmentation. Other graph-based methods [38, 30, 35] and transformer-based methods [19, 40] share the similar high-level idea to convey the category information from the support set to the query image.

However, these methods all rely heavily on the category information extracted from the support set. Although it provides deterministic category information guidance, there may exist inherent

---

[*]Equal contribution.
[†]Corresponding author.

36th Conference on Neural Information Processing Systems (NeurIPS 2022).

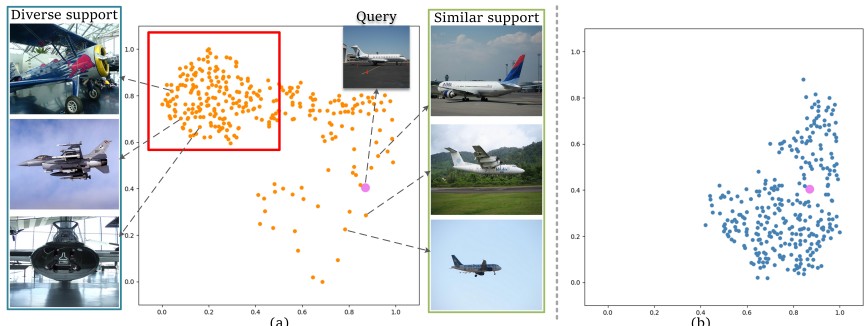

Figure 1: t-SNE visualization of the prototype distribution. (a): The distribution of support and query prototypes. (b): The distribution of intermediate and query prototypes. Our proposed intermediate prototypes are closer to the query than the support prototypes.

intra-class diversity between query and support samples, which is collectively ignore by existing works. In Figure 1 (a), we show the distribution of some support prototypes (in orange) and the prototype generated from a query image (in magenta) of the airplane class. We observe that for the support images that are similar with the query image (marked as "Similar support" on the right), their prototypes are close to the query prototype in the feature space, in which case the matching network can work well. However, for the support images that have large diversity in pose and appearance compared with the query (marked as "Diverse support" on the left), the distance between the support and query prototypes will be faraway. In such a case, if we forcibly migrate the category information in the support prototype to the query, a large category information bias is inevitably introduced.

Therefore, our work aims to relieve this problem by introducing an intermediate prototype that could bridge the category information gap between query and support images through our proposed Intermediate Prototype Mining Transformer (IPMT). Each IPMT layer consists of two steps, *i.e.*, Intermediate Prototype Mining (IPM) and Query Activation (QA). In IPM we learn the intermediate prototype via combing both the deterministic category information from the support images and the adaptive category knowledge from the query image. Then, we use the learned prototype to activate the query feature map in QA. Furthermore, our IPMT is used in an iterative way to progressively improve the quality of the learned prototype and the activated query feature. As such, the intermediate prototype can successfully reduce the category information gap with the query set, which is verified by the distribution in Figure 1 (b).

In summary, our main contributions can be concluded as: 1) To the best of our knowledge, this is the first time to focus on the intra-class diversity between support and query in FSS, and we propose the idea of intermediate prototype to relieve the existing category information gap issue. 2) We propose a novel IPMT to explicitly mine the intermediate prototype which contains both the deterministic information from the support set and the adaptive category knowledge from the query. 3) We present an iterative learning scheme to fully explore the intermediate category information hidden in both support and query and update the query feature. 4) Extensive experiments on PASCAL-$5^i$ and COCO-$20^i$ show that our proposed IPMT brings a significant improvement over state-of-the-art methods.

## 2 Related Work

### 2.1 Visual Transformer

Recently, transformer is introduced into the field of artificial intelligence and has attracted significant attention from many researchers. [29] first proposes the transformer architecture composed of self-attention and feed-forward layers, and also achieves remarkable performance in natural language processing. Very recently, transformer has been found to be able to obtain excellent results in computer vision. Specifically, [5] firstly introduces transformer into vision tasks and proposes vision transformer (ViT) by treating non-overlapped image patches as a series of tokens. Subsequently, a lot of works are devoted to tailoring the transformer structure to be more suitable for computing vision tasks. [18] utilizes shifted window self-attention and patch merging to reduce the computational cost and aggregate features, respectively. [33] imitates the feature pyramid structure in CNNs and proposes the pyramid vision transformer, which reduces the computational cost of self-attention.

Other transformer-based methods are proposed for various tasks, such as object detection[42, 3], semantic segmentation[41], panoptic segmentation[31], multiple object tracking[27] and so on. [3] proposes an transformer-based end-to-end object detection framework and utilizes object queries to locate the objects. [7] replaces the multi-head attention with twin attention to interact with the context both on row and column features for instance segmentation. In the above works, learnable Queries* are utilized to aggregate the context-information from feature maps for further regression or segmentation. [4] introduces the masked attention to extract relevant features instead of global features by limiting the cross-attention regions within predicted masks. Motivated by these works, we use a learnable prototype as the intermediate prototype to dig out adaptive category information from both query and support images.

## 2.2 Few-shot Semantic Segmentation

FSS is a natural extension of semantic segmentation in the condition of a few annotated samples. The typical paradigm proposed by SG-one [25] is using two-branch networks. A conditioning branch extracts the category context from the support images and another segmentation branch segments the query image under the guidance of the former. Following this paradigm, many approaches [36, 12, 37, 14, 28, 34] are proposed to explore how to fully excavate the category information from the support images. For mining more abundant category information from the support images, ASGNet[36] and PMM[12] construct multiple prototypes using parameter-free methods, i.e. superpixel-guided clustering and the expectation-maximum algorithm, to cluster the support foreground features into multiple prototypes and then activate different areas in the query image. SCL[37] utilizes the missing parts in the initial segmentation result of the support images to form auxiliary support vectors and then merge them in a cross-guidance module to obtain a better prediction. In PFENet[28], a prior mask is generated by calculating the cosine-similarity between support and query features in high-level. Then, a feature enrichment module is applied to perform dense comparison on different feature scales obtained by adaptive pooling. Furthermore, MMNet[34] introduces meta-class memory to store the meta-information during training and applies it into novel classes during the inference stage. However, all above methods ignore the inherent intra-class differences between query and support images and transfer the support information to the query image forcibly. Our work aims to relieve this problem and propose the intermediate prototype to bridge the category information gap between query and support images.

## 2.3 Transformer-based Few-shot Semantic Segmentation

[19] introduces the multi-head attention as an attention module to transfer the classifier weights from support to query. However, it does not take full advantage of the transformer on incorporating long-range dependencies. [40] proposes a Cycle-Consistent TRansformer (CyCTR) module to select relevant pixel-level support features to perform cross-attention with the query feature. In our work, instead of performing cross-attention between support and query features, we leverage a learnable Query as the intermediate prototype to aggregate the category information from both support and query images and refine the query feature using this prototype.

## 3 Problem Definition

In FSS, the whole dataset is divided into two disjoint subsets $\mathcal{D}_{train}$ and $\mathcal{D}_{test}$ based on the object categories they contain. An FSS model is expected to learn the meta knowledge on $\mathcal{D}_{train}$ with sufficient labeled images and generalize to unseen categories on $\mathcal{D}_{test}$ with scarce labeled images. Following the previous meta-learning paradigm, we execute the episodic training strategy to train our model. Specifically, these two subsets are both partitioned into numerous episodes, each of which randomly samples $K + 1$ image-mask pairs. For one episode, $K$ pairs compose the support set $\mathcal{S} = \{(\mathbf{I}_i^s, \mathbf{M}_i^s)\}_{i=1}^K$ and the rest one pair composes the query set $\mathcal{Q} = \{(\mathbf{I}^q, \mathbf{M}^q)\}$, where $\mathbf{I}^* \in \mathbb{R}^{H \times W \times 3}$ and $\mathbf{M}^* \in \mathbb{R}^{H \times W}$ denote the RGB images and their corresponding binary masks, respectively. In each episode sampled from $\mathcal{D}_{train}$, the model is trained to predict the query mask supervised by $\mathbf{M}^q$ under the guidance of the support set. Then, the trained model is evaluated on $\mathcal{D}_{test}$ to segment unseen categories straightly without any further optimization.

---

*Here, we use the initially capitalized 'Query' and the lowercase 'query' to distinguish the context of query in transformer and FSS respectively.

# 4   A Review of Transformer

We first review the typical transformer model here. As the form in [29], a transformer layer mainly consists of an attention block and a multi-layer perception (MLP) block. The former is used to aggregate global contexts and the latter performs embedding updating.

**Attention Block.**   Given an input token sequence $\mathbf{X} \in \mathbb{R}^{L_1 \times C}$ and a context token sequence $\mathbf{Y} \in \mathbb{R}^{L_2 \times C}$, where $L_1, L_2$ are the length of the two sequences and $C$ is the channel dimension of their embeddings, respectively, the attention block first computes the attention weight matrix:

$$\mathbf{A}(\mathbf{X}, \mathbf{Y}) = \frac{\mathbf{X}\mathbf{W_q}(\mathbf{Y}\mathbf{W_k})^\top}{\sqrt{d}}, \tag{1}$$

where $\mathbf{W_q}$ and $\mathbf{W_k} \in \mathbb{R}^{C \times d}$ are linear transformation weight matrixes and $\sqrt{d}$ is the scale factor. Then, $\mathbf{A}(\mathbf{X}, \mathbf{Y})$ is normalized and then used to aggregate the global context from $\mathbf{Y}$:

$$\mathbf{Attn}(\mathbf{X}, \mathbf{Y}) = Softmax(\mathbf{A}(\mathbf{X}, \mathbf{Y}))\mathbf{Y}\mathbf{W_v}, \tag{2}$$

where $\mathbf{W}_v \in \mathbb{R}^{C \times C}$ is another linear transformation weight matrix.

If $\mathbf{X}, \mathbf{Y}$ are the same feature, the attention operation is called self-attention which propagates contexts among different tokens. If they are not the same, it is named cross-attention which conveys relevant information from $\mathbf{Y}$ to $\mathbf{X}$. Usually multi-head attention [29] is used to boost the model performance.

**Multi-layer Perception Block.**   After the attention block, a MLP block is applied to each token separately and identically to further transform the token embeddings. Specifically, MLP is implemented using two linear projection layers with a ReLU activation in between. Given a token sequence $\mathbf{X} \in \mathbb{R}^{L_1 \times C}$ as the input, it is formulated as:

$$\mathbf{MLP}(\mathbf{X}) = ReLU(\mathbf{X}\mathbf{W_1} + \mathbf{b_1})\mathbf{W_2} + \mathbf{b_2}, \tag{3}$$

where $\mathbf{W}_*$ and $\mathbf{b}_*$ denote the linear transformation weight matrixes and biases, respectively. We follow [29] and set the channel dimensions of the first and the second layer to be $4C$ and $C$, respectively. Note that layer normalization [1] and residual connections are omitted here for simplicity.

# 5   Intermediate Prototype Mining Transformer

We now present our proposed Intermediate Prototype Mining Transformer (IPMT), as Figure 2 shown, for few-shot semantic segmentation. Each IPMT layer consists of two steps, *i.e.*, Intermediate Prototype Mining (IPM) and Query Activation (QA). IPM is used to mine the intermediate prototype from both support and query features while QA is designed to activate the query feature map using the learned prototype. We adopt a duplex segmentation loss (DSL) to supervise the learning of the intermediate prototype in each IPMT layer. Furthermore, we propose to perform the intermediate prototype mining in an iterative way, thus boosting the quality of the learned prototype and the segmentation results progressively. Next, we will describe them in details.

## 5.1   Intermediate Prototype Mining

Our IPM has a learnable prototype $\mathbf{G}$ to extract adaptive category information from both query and support images using Masked Attention (MA). The prototype $\mathbf{G} \in \mathbb{R}^{1 \times C}$ is initially a category- and image-agnostic vector that encodes general segmentation prior and will be updated by MA in each episode, encoding adaptive category information for the target category in that episode.

**Masked Attention.**   We leverage cross-attention to update $\mathbf{G}$ by using both support and query features as the context. Furthermore, to make $\mathbf{G}$ only focus on the target regions and extract the category information without noises, we follow [4] and use support and query masks to limit the attended region in the attention matrix. Specifically, given a flattened support or query feature $\mathbf{F} \in \mathbb{R}^{hw \times C}$ and a corresponding binary segmentation mask $\mathbf{P} \in \mathbb{R}^{h \times w}$, we first compute the attention weight matrix $\mathbf{A}(\mathbf{G}, \mathbf{F}) \in \mathbb{R}^{1 \times hw}$. Then, an attention mask is computed by following [4]:

$$\hat{\mathbf{P}}(i) = \left\{ \begin{array}{cc} 0 & \text{if } \mathbf{P}(i)\text{=1} \\ -\infty & \text{otherwise} \end{array} \right. , \tag{4}$$

where $i$ denotes the location index. Next, we use $\hat{\mathbf{P}}$ to modulate the attention weights, leading to the masked attention:

$$\mathbf{MaskAttn}(\mathbf{G}, \mathbf{F}, \mathbf{P}) = Softmax(\mathbf{A}(\mathbf{G}, \mathbf{F}) + \mathbf{Vec}(\hat{\mathbf{P}}))\mathbf{F}\mathbf{W_v}, \tag{5}$$

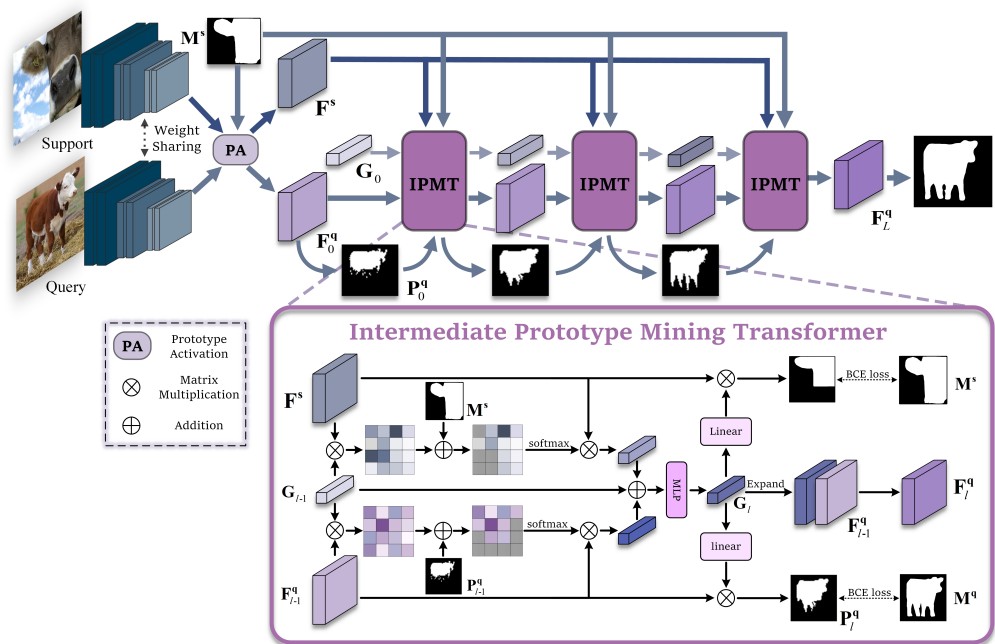

Figure 2: Overall architecture of our proposed IPMT. Support and query images are first fed into two pre-trained backbone encoders to extract features. Then, we follow previous works [28, 40] and conduct prototype activation (PA) to obtain the support and initial query features $\mathbf{F^s}$ and $\mathbf{F_0^q}$, respectively. Meanwhile, the initial query segmentation mask $\mathbf{P_0^q}$ is obtained from $\mathbf{F_0^q}$. Next, we feed $\mathbf{F^s}$, $\mathbf{F_0^q}$, $\mathbf{P_0^q}$, and the support mask $\mathbf{M}^s$, the initial intermediate prototype $\mathbf{G}_0$ into our IPMT layers to iteratively update the prototype, the query mask, and the query feature. After $L$ iterations, the final query feature $\mathbf{F}_L^q$ is used to obtain the final query segmentation result.

where $\mathbf{Vec}(\cdot)$ is the vectorization operation. As such, the normalized attention weights on the background regions are close to zero, making sure the prototype only be updated by relevant contexts of the desired category.

In our IPM, on one hand, we use the support feature $\mathbf{F^s}$ and the ground-truth support mask $\mathbf{M^s}$ to update $\mathbf{G}$, providing deterministic category information since $\mathbf{M^s}$ is definitely accurate. On the other hand, we also leverage the query feature $\mathbf{F^q}$ and a query prediction mask $\mathbf{P^q}$ to provide query-adaptive category knowledge for $\mathbf{G}$, thus reducing the category information gap between support and query images. After that, an MLP block is further used on the learned prototype. The whole process can be formulated as:

$$\mathbf{IPM}(\mathbf{G}, \mathbf{F^s}, \mathbf{F^q}, \mathbf{M^s}, \mathbf{P^q}) = \mathbf{MLP}(\mathbf{MaskAttn}(\mathbf{G}, \mathbf{F^s}, \mathbf{M^s}) + \mathbf{MaskAttn}(\mathbf{G}, \mathbf{F^q}, \mathbf{P^q}) + \mathbf{G}). \quad (6)$$

Please note that the two masked attention operations do *not* share weights since the two segmentation masks have different uncertainty. For the $K$-shot setting, we simply average the outputs of $\mathbf{MaskAttn}$ on support.

## 5.2 Query Activation

In this step, QA is used to activate the target regions in the query feature map $\mathbf{F^q}$ under the guidance of the learned prototype $\mathbf{G}$. Previous works [28, 39] have demonstrated that it is an essential operation for FSS to pass the category information to the query feature map and provide specific segmentation cues. Specifically, $\mathbf{G}$ is expanded and concatenated with $\mathbf{F^q} \in \mathbb{R}^{h \times w \times C}$ to activate the target regions:

$$\mathbf{QA}(\mathbf{G}, \mathbf{F^q}) = \mathcal{F}_{actv}(\mathbf{G} \odot \mathbf{F^q}), \quad (7)$$

where $\odot$ represents the concatenation operation, and $\mathcal{F}_{actv}$ is a simple activation network which consists of a $1 \times 1$ convolutional layer, a ReLU layer, and a $3 \times 3$ convolutional layer. Additionally, we also follow [40] to use a multi-head deformable self-attention layer [42] for further aggregating context information in the query feature.

## 5.3 Duplex Segmentation Loss

To facilitate the learning of the adaptive category information in $\mathbf{G}$, we use it to generate two segmentation masks on both support and query images and calculate two segmentation losses. Specifically, motivated by [4], we use $\mathbf{G}$ to generate a mask embedding and then conduct multiplication with the image feature maps for obtaining segmentation masks. The mask generation (MG) process is formulated as:

$$\mathbf{MG}(\mathbf{G}, \mathbf{F^q}) = Sigmoid(\mathbf{G}\mathbf{W_m}(\mathbf{F^q})^\top), \tag{8}$$

$$\mathbf{MG}(\mathbf{G}, \mathbf{F^s}) = Sigmoid(\mathbf{G}\mathbf{W_m}(\mathbf{F^s})^\top), \tag{9}$$

where $\mathbf{W_m} \in \mathbb{R}^{C \times C}$ is a linear projection weight matrix for generating the mask embedding.

Next, the standard binary cross-entropy (BCE) loss is calculated between the generated masks and the ground truth, *i.e.*, $\mathbf{M^q}$ and $\mathbf{M^s}$, as our duplex segmentation loss to optimize the prototype learning process:

$$\mathcal{L}^{dsl} = \alpha BCE(\mathbf{MG}(\mathbf{G}, \mathbf{F^q}), \mathbf{M^q}) + (1 - \alpha)BCE(\mathbf{MG}(\mathbf{G}, \mathbf{F^s}), \mathbf{M^s}). \tag{10}$$

Here, $\alpha$ is a hyperparameter to balance the losses between query and support predictions.

## 5.4 Iterative Prototype Mining

Since one IPMT layer can update the intermediate prototype $\mathbf{G}$, the query feature map $\mathbf{F^q}$, and the query segmentation mask $\mathbf{P^q}$, we can iteratively perform this process and obtain better and better $\mathbf{G}$ and $\mathbf{F^q}$, finally making the segmentation results effectively boosted. Suppose we have $L$ iterative IPMT layers, then for each layer $l$ we have:

$$\mathbf{G}_l, \mathbf{F}_l^\mathbf{q}, \mathbf{P}_l^\mathbf{q} = \mathbf{IPMT}(\mathbf{G}_{l-1}, \mathbf{F^s}, \mathbf{F}_{l-1}^\mathbf{q}, \mathbf{M^s}, \mathbf{P}_{l-1}^\mathbf{q}), \tag{11}$$

which can be broken down into the following steps:

$$\mathbf{G}_l = \mathbf{IPM}(\mathbf{G}_{l-1}, \mathbf{F^s}, \mathbf{F}_{l-1}^\mathbf{q}, \mathbf{M^s}, \mathbf{P}_{l-1}^\mathbf{q}), \tag{12}$$

$$\mathbf{F}_l^\mathbf{q} = \mathbf{QA}(\mathbf{G}_l, \mathbf{F}_{l-1}^\mathbf{q}), \tag{13}$$

$$\mathbf{P}_l^\mathbf{q} = \mathbf{MG}(\mathbf{G}_l, \mathbf{F}_{l-1}^\mathbf{q}) \geq 0.5. \tag{14}$$

Here, since the masked attention requires a binary mask as the input, we use 0.5 as the threshold to generate $\mathbf{P}_l^\mathbf{q}$.

# 6 Few-shot Semantic Segmentation Model

Following previous works [28, 40], we input query and support images into a fixed and shared encoder backbone such as the ResNet family [9] to obtain multi-level features. Then, we concatenate the outputs of the third and fourth encoder blocks together and then adopt a $1 \times 1$ convolutional layer to generate middle-level query and support features, respectively. We also calculate the similarity between the high-level query and support features at the fifth encoder block to produce a prior mask and use masked average pooling on the support feature map to obtain a support prototype. Next, the query feature map, prior mask, and the expanded prototype are concatenated and transformed using a $1 \times 1$ convolutional layer to obtain the initial query feature $\mathbf{F}_0^\mathbf{q}$. We also concatenate the middle-level support feature with the expanded prototype to generate the support feature $\mathbf{F^s}$. All the above processes are common methods in FSS and termed prototype activation (PA) in Figure 2. For more details please refer to [28].

As for the iterative learning, we feed $\mathbf{F}_0^\mathbf{q}$ into two convolutional layers to obtain the initial query segmentation prediction $\mathbf{P}_0^\mathbf{q}$. The initial intermediate prototype $\mathbf{G}_0$ is randomly initialized at the beginning of the training and then optimized on the training set. Next, we feed $\mathbf{G}_0, \mathbf{F^s}, \mathbf{F}_0^\mathbf{q}, \mathbf{M^s}$, and $\mathbf{P}_0^\mathbf{q}$ into our iterative IPMT layers to perform intermediate prototype mining. After $L$ iterations, the final activated query feature $\mathbf{F}_L^\mathbf{q}$ is used to predict the final segmentation result via two convolutional layers. The dice loss [20] is used here to optimize the training of the whole model.

Table 1: Class mIoU results of four folds on PASCAL-$5^i$. The results of 'Mean' are the averaged class mIoU scores of all four folds. **Red**/**Blue** indicates the best/$2^{nd}$ results.

| Backbone | Methods | 1-shot | | | | | 5-shot | | | | |
|---|---|---|---|---|---|---|---|---|---|---|---|
| | | Fold-0 | Fold-1 | Fold-2 | Fold-3 | **Mean** | Fold-0 | Fold-1 | Fold-2 | Fold-3 | **Mean** |
| ResNet-50 | RPMMs(ECCV'20)[36] | 55.2 | 66.9 | 52.6 | 50.7 | 56.3 | 56.3 | 67.3 | 54.5 | 51.0 | 57.3 |
| | PFENet(TPAMI'20) | 61.7 | 69.5 | 55.4 | 56.3 | 60.8 | 63.1 | 70.7 | 55.8 | 57.9 | 61.9 |
| | RePRI(CVPR'21)[2] | 59.8 | 68.3 | **62.1** | 48.5 | 59.7 | 64.6 | 71.4 | **71.1** | 59.3 | 66.6 |
| | HSNet(ICCV'21)[21] | 64.3 | 70.7 | 60.3 | **60.5** | 64.0 | **70.3** | 73.2 | 67.4 | **67.1** | **69.5** |
| | CWT(ICCV'21)[19] | 56.3 | 62.0 | 59.9 | 47.2 | 56.4 | 61.3 | 68.5 | **68.5** | 56.6 | 63.7 |
| | CyCTR(NeurIPS'21)[40] | **65.7** | 71.0 | 59.5 | 59.7 | 64.0 | 69.3 | **73.5** | 63.8 | 63.5 | 67.5 |
| | NERTNet(CVPR'22)[16] | 65.4 | **72.3** | 59.4 | 59.8 | **64.2** | 66.2 | 72.8 | 61.7 | 62.2 | 65.7 |
| | DCP(IJCAI'22)[11] | 63.8 | 70.5 | **61.2** | 55.7 | 62.8 | 67.2 | 73.1 | 66.4 | **64.5** | 67.8 |
| | IPMT(ours) | **72.8** | **73.7** | 59.2 | **61.6** | **66.8** | **73.1** | **74.7** | 61.6 | 63.4 | **68.2** |
| ResNet-101 | DAN(ECCV'20)[30] | 54.7 | 68.6 | 57.8 | 51.6 | 58.2 | 57.9 | 69.0 | 60.1 | 54.9 | 60.5 |
| | PFENet(TPAMI'20)[28] | 60.5 | 69.4 | 54.4 | 55.9 | 60.1 | 62.8 | 70.4 | 54.9 | 57.6 | 61.4 |
| | CWT(ICCV'21)[19] | 56.9 | 65.2 | **61.2** | 48.8 | 58.0 | 62.6 | 70.2 | **68.8** | 57.2 | 64.7 |
| | NERTNet(CVPR'22)[16] | 65.5 | 71.8 | **59.1** | 58.3 | 63.7 | 67.9 | 73.2 | **60.1** | **66.8** | **67.0** |
| | CyCTR(NeurIPS'21)[40] | **69.3** | **72.7** | 56.5 | **58.6** | **64.3** | **73.5** | **74.0** | 58.6 | 60.2 | 66.6 |
| | IPMT(ours) | **71.6** | **73.5** | 58.0 | **61.2** | **66.1** | **75.3** | **76.9** | 59.6 | **65.1** | **69.2** |

# 7 Experiments

## 7.1 Datasets and Evaluation Metrics

**Datasets.** To make a fair comparison with previous works, our model is evaluated on two few-shot semantic segmentation benchmark datasets, *i.e.*, the PASCAL-$5^i$ dataset[25] and the COCO-$20^i$ dataset [22]. PASCAL-$5^i$ is constructed based on the PASCAL VOC 2012 dataset [6] and additional annotations from SDS [8]. It contains 20 categories in total and these categories are partitioned into four folds as in [32] for cross validation, where each fold contains five categories. COCO-$20^i$ is a larger datasets based on the MSCOCO [13] dataset. Similar to the division in PASCAL-$5^i$, the 80 categories in MSCOCO are also partitioned into four folds for cross validation, where each fold includes 20 categories. For both datasets, we train our model on three folds and evaluate it on the remaining one fold and perform cross validation.

**Evaluation Metrics.** Following previous methods [25, 26, 15, 17], we adopt the class mean intersection over union (mIoU) as a primary evaluation metric. In addition, we also report the results of foreground-background IoU (FB-IoU) for comparison.

## 7.2 Implementation Details

Following previous works, we adopt ResNet-50 and ResNet-101 [9] as our encoder backbone. Note that they are initialized by the weights pre-trained on ImageNet [24] and froze during training.

Our proposed IPMT is implemented using PyTorch [23] and all the experiments are conducted on one NVIDIA RTX 3090 GPU. We use the same data augmentation setting as [28] for fair comparisons. Our model is trained for 200 epochs on PASCAL-$5^i$ while 50 epochs on COCO-$20^i$, respectively, with the batchsize set to 4. Two optimizers (*i.e.*, SGD and AdamW) are used to train our model. The former is used to optimize the convolutional layers with the initial learning rate, weight decay, and momentum set to $2.5 \times 10^{-3}$, 0.0001, and 0.9, respectively. The latter is used to optimize the transformer layers by setting the learning rate to $1 \times 10^{-4}$ and the weight decay to $1 \times 10^{-2}$. We also use the polynomial annealing policy with the power set to 0.9 to decay the learning rate. For hyper-parameters in our IPMT, the number of multi-heads and the channel dimension of the image features are set to 8 and 256, respectively. The weight $\alpha$ in DSL is set to 0.3 since a larger weight should be given for the more reliable category knowledge from the support images. During the evaluation, we follow [28] to randomly sample 1000 support-query pairs on PASCAL-$5^i$ and 4000 pairs on COCO-$20^i$, respectively.

## 7.3 Comparison with State-of-the-art Methods

**Quantitative comparison.** As shown in Table 1 and 2, we compare our method with previous works on both PASCAL-$5^i$ and COCO-$20^i$, respectively. It can be found that our IPMT surpasses all other approaches by a large margin and achieves new state-of-the-art results. On PASCAL-$5^i$, when using ResNet-50 as the backbone, our proposed IPMT improves the mIoU score by 2.6 under the

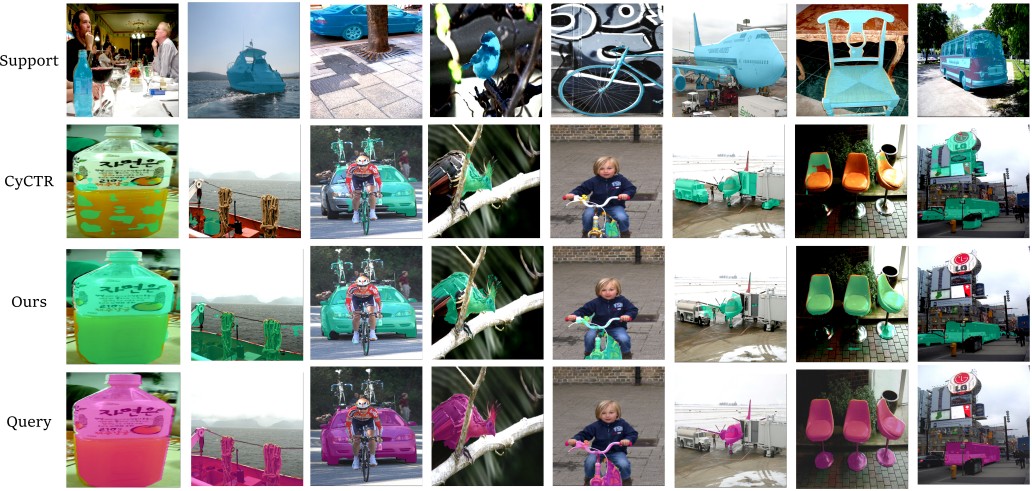

Figure 3: Qualitative comparison between our proposed IPMT and CyCTR [40]. From top to bottom: support images, prediction of CyCTR, prediction of IPMT, query images.

Table 2: Class mIoU results of four folds on COCO-$20^i$. The results of 'Mean' are the averaged class mIoU scores of all the four folds. **Red**/**Blue** indicates the best/$2^{nd}$ results.

| Backbone | Methods | 1-shot | | | | | 5-shot | | | | |
|---|---|---|---|---|---|---|---|---|---|---|---|
| | | Fold-0 | Fold-1 | Fold-2 | Fold-3 | **Mean** | Fold-0 | Fold-1 | Fold-2 | Fold-3 | **Mean** |
| ResNet-50 | RPMMs(ECCV'20)[36] | 29.5 | 36.8 | 29.0 | 27.0 | 30.6 | 33.8 | 42.0 | 33.0 | 33.3 | 35.5 |
| | RePRI(CVPR'21)[2] | 31.2 | 38.1 | 33.3 | 33.0 | 34.0 | 38.5 | 46.2 | 40.0 | 43.6 | 42.1 |
| | HSNet(ICCV'21)[21] | 36.3 | 43.1 | 38.7 | 38.7 | 39.2 | 43.3 | **51.3** | **48.2** | 45.0 | **46.9** |
| | CWT(ICCV'21)[19] | 32.2 | 36.0 | 31.6 | 31.6 | 32.9 | 40.1 | 43.8 | 39.0 | 42.4 | 41.3 |
| | CyCTR(NeurIPS'21)[40] | 38.9 | 43.0 | 39.6 | **39.8** | 40.3 | 41.1 | 48.9 | 45.2 | **47.0** | 45.6 |
| | DCP(IJCAI'22)[11] | **40.9** | **43.8** | **42.6** | 38.3 | **41.4** | **45.8** | 49.6 | 43.7 | 46.6 | 46.5 |
| | NERTNet(CVPR'22)[16] | 36.8 | 42.6 | 39.9 | 37.9 | 39.3 | 38.2 | 44.1 | 40.4 | 38.4 | 40.3 |
| | IPMT(ours) | **41.4** | **45.1** | **45.6** | **40.0** | **43.0** | **43.5** | 49.7 | 48.7 | **47.9** | **47.5** |
| ResNet-101 | PFENet(TPAMI'20)[28] | 34.3 | 33.0 | 32.3 | 30.1 | 32.4 | 38.5 | 38.6 | 38.2 | 34.3 | 37.4 |
| | CWT(ICCV'21)[19] | 30.3 | 36.6 | 30.5 | 32.2 | 32.4 | 38.5 | 46.7 | 39.4 | **43.2** | 42.0 |
| | SCL(CVPR'21)[37] | 36.4 | 38.6 | 37.5 | 35.4 | 37.0 | 38.9 | 40.5 | 41.5 | 38.7 | 39.9 |
| | SAGNN(CVPR'21)[35] | 36.1 | **41.0** | 38.2 | 33.5 | 37.2 | 40.9 | **48.3** | 42.6 | 38.9 | 42.7 |
| | NERTNet(CVPR'22)[16] | **38.3** | 40.4 | **39.5** | **38.1** | **39.1** | **42.3** | 44.4 | **44.2** | 41.7 | **43.2** |
| | IPMT(ours) | **40.5** | **45.7** | **44.8** | **39.3** | **42.6** | **45.1** | **50.3** | **49.3** | **46.8** | **47.9** |

1-shot setting compared with the previous best result. Additionally, we also achieve 1.8 mIoU (under the 1-shot setting) and 2.2 mIoU (5-shot setting) improvements using the ResNet-101 backbone. As for COCO-$20^i$, our method with the ResNet-50 backbone outperforms the previous best results by 1.6 and 0.6 mIoU under the two settings, respectively. When using the ResNet-101 backbone, our IPMT improves the mIoU score by 3.5 and 4.7 over the previous best results. These improvements demonstrate the competitiveness of our model on complex data. In addition, we report the comparison with some advanced approaches in terms of the FB-IoU score in Table 3, which also shows our superiority.

**Quantitative Result.** We visualize some prediction results of our method and a support-only FSS method (*i.e.*, CyCTR [40]) in Figure 3. It can be observed that our results (the 3 row) could effectively relieve the segmentation error caused by the inherent intra-class diversity compared with the results of only using the support information (the 2th row).

## 7.4 Ablation Study

In this section, we report ablation study results on PASCAL-$5^i$ with the ResNet-50 backbone under the 1-shot setting.

**Effectiveness of IPM.** To demonstrate the necessity of bridging the category information gap between query and support images using the proposed IPM, we conduct ablation study by learning the category information only from support or query images. Furthermore, we report the performance comparison both with and without iteration. All the ablation studies are conducted with both DSL and

Table 3: Comparison with state-of-the-arts on PASCAL-$5^i$ in terms of FB-IoU under the 1-shot and 5-shot settings.

| Backbone | Methods | FB-IoU | |
| --- | --- | --- | --- |
| | | 1-shot | 5-shot |
| ResNet-50 | PPNet(ECCV'20)[17] | 69.2 | 75.7 |
| | PFENet(TPAMI'20)[28] | 73.3 | 73.9 |
| | HSNet(ICCV'21)[21] | **76.7** | **80.6** |
| | SCL(CVPR'21)[37] | 71.9 | 72.8 |
| | DCP(IJCAI'22)[11] | 75.6 | 79.7 |
| | IPMT(ours) | **77.1** | **81.4** |
| ResNet-101 | A-MCG(AAAI'19)[10] | 61.2 | 62.2 |
| | DAN(ECCV'20) [30] | 71.9 | 72.3 |
| | PFENet(TPAMI'20)[28] | 72.9 | 73.5 |
| | CyCTR(NeurIPS'21)[40] | **73.0** | **75.4** |
| | IPMT(ours) | **78.5** | **80.3** |

Table 4: Ablation study on the effectiveness of IPM.

| Support only | Query only | Intermediate | Iteration | mIoU |
| --- | --- | --- | --- | --- |
| ✓ | | | | 62.5 |
| | ✓ | | | 59.8 |
| | | ✓ | | 64.1 |
| ✓ | | | ✓ | 63.4 |
| | ✓ | | ✓ | 60.1 |
| | | ✓ | ✓ | **66.8** |

Table 5: Performance comparison of varying the number of IPMT layers.

| Layers | 1 | 2 | 3 | 4 | 5 |
| --- | --- | --- | --- | --- | --- |
| mIoU | 64.1 | 64.7 | 65.2 | 65.6 | **66.8** |

Table 6: Ablation study on the effectiveness of DSL and QA.

| IPM | DSL | QA | mIoU |
| --- | --- | --- | --- |
| | | | 60.2 |
| ✓ | | | 54.9 |
| ✓ | ✓ | | 64.3 |
| ✓ | ✓ | ✓ | **66.8** |

Table 7: Intra-class diversity measured by Euclidean distances among the query, support, and intermediate prototypes of four folds on PASCAL-$5^i$.

| | fold-0 | fold-1 | fold-2 | fold-3 | Mean |
| --- | --- | --- | --- | --- | --- |
| $D_{qs}$ | 7.624 | 7.784 | 6.875 | 9.430 | 7.928 |
| $D_{qi}$ | 6.905 | 6.775 | 5.941 | 8.249 | 6.968 |
| $D_{is}$ | 3.616 | 3.994 | 3.520 | 6.202 | 4.333 |

QA for a fair comparison. As shown in Table 4, 'Support only' and 'Query only' mean the learnable prototype is only updated by the support feature and the query feature, respectively. 'Intermediate' means using both support and query information in our IPM. We observe that our method surpasses other schemes by 1.6 mIoU when not using iteration. This improvement even increases to 3.4 when using iteration. These results clearly demonstrate the effectiveness of our IPM and also indicate that using iteration is an effective booster for our IPM.

**Ablation on Different Numbers of IPMT Layers.** We vary the number of IPMT layers from 1 to 5 and report the results in Table 5. It shows that using more layers can gradually improve the model performance and using five layers achieves as much as 2.7 mIoU improvement. We did not try more layers considering the accuracy-efficiency trade-off.

**Effectiveness of DSL and QA.** Since the effectiveness of the IPM has been proved in Table 4, here we conduct an ablation study to validate the effectiveness of QA and DSL. We remove all the three components from our model as the baseline model (only with PA), which only uses the support prototype to directly segment the target object like [28]. As Table 6 shows, compared to the baseline, solely using IPM leads to 5.3 mIoU drop. However, when DSL is added, our model achieves 4.1 mIoU improvement over the baseline. This phenomenon is reasonable because there is no guarantee that the learnable prototype in IPM will learn intermediate category knowledge without DSL. Meanwhile, using the QA to activate the query feature map leads to further 2.5 mIoU improvement. These results clearly verify the effectiveness of our proposed QA and DSL.

**Prototype Comparison.** We first visualize the overall distribution of the support (orange points) and intermediate (blue points) prototypes given two query (magenta points) images in Figure 4 (a) and (c). It is clearly observed that our intermediate prototypes are closer to the query prototypes than the support ones are in the feature space, hence verifying that our method effectively relieves the intra-class diversity issue and bridges the category information gap between query and support images. We also show the support images for some specific points and the query images in Figure 4 (b) and (d). Please note that the support and intermediate prototypes of the same support image are shown as two points with the same shape. We find that for these support images which have clear intra-class differences from the query image, our generated intermediate prototypes are successfully pulled to a closer feature space with the query prototype.

Additionally, for evaluating the intra-class diversity objectively, we adopt the Euclidean distance as a metric to measure the distance between the query prototype and the support prototype ($D_{qs}$) in each episode. Then, to further demonstrate the effectiveness of our method, we also measure the distance between the query prototype and the intermediate prototype ($D_{qi}$) and the distance between the intermediate prototype and the support prototype ($D_{is}$). The average distances of each fold and the mean of all the categories on the PASCAL-$5^i$ are shown in Table 7. From the table, we can clearly see that $D_{qi}$ is smaller than $D_{qs}$ on all folds, which means that our mined intermediate prototype is more similar to the query than the support is. This also demonstrates that our method effectively reduces the distance between the mined prototype and the query and mitigates the intra-class diversity problem.

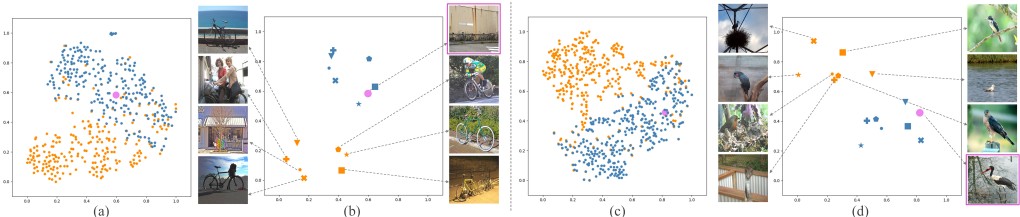

Figure 4: Comparison of the distribution of support and intermediate prototypes. (a) & (c): The overall distribution of support prototypes and intermediate prototypes. The latter are closer to the query than the former. (b) & (d): The visualization of the query images and the support images of some points. Different marks indicate different support prototypes and their corresponding intermediate prototypes.

## 8 Conclusion

In this paper, we focus on the intra-class diversity between query and support and introduce an intermediate prototype to bridge the category information gap between them. The core idea is to use the intermediate prototype to aggregate the support-deterministic and query-adaptive category information by our designed Intermediate Prototype Mining Transformer (IPMT) in an iterative way. Surprisingly, despite its simplicity, our method outperforms previous state-of-the-art results by a large margin on two FSS benchmark datasets. We hope our work could inspire future research to concentrate more on the intra-class diversity in FSS.

## Acknowledgments

This work was supported in part by the National Key RD Program of China under Grant 2020AAA0105701, the National Natural Science Foundation of China under Grant 62136007, 62071388, 62036011 and U20B2065, the Key Research and Development Program of Shaanxi Province under Grant 2021ZDLGY01-08.

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
