# Supplementary to "Intermediate Prototype Mining Transformer for Few-Shot Semantic Segmentation"

**Yuanwei Liu**[1]*    **Nian Liu**[2]*    **Xiwen Yao**[1]    **Junwei Han**[1]†
[1]Northwestern Polytechnical University
[2]Mohamed bin Zayed University of Artificial Intelligence
{liuyuanwei9809, liunian228, yaoxiwen517, junweihan2010}@gmail.com

## 1    Appendix

### 1.1    Limitation and Societal Impacts

**Limitation.**    We observe that the performance of our method does not achieve a significant improvement under the 5-shot setting. We argue that this is reasonable because as the number of the support images increases, their internal diversity probably increases. As a result, the intra-class diversity between support and query images probably decreases and their category information gap will be narrowed. Thus, the ability of our method to correct the support-query deviation will be weakened. However, our work still performs very effectively under the 1-shot setting and provides a new perspective for future research.

**Societal Impacts.**    The method proposed in this paper is applied for few-shot semantic segmentation which has many applications in robot vision systems, image recognition software, etc. Compared with previous methods, although our method achieves a large performance improvement, it is still hard to come up with the human cognitive ability. For some safety-critical fields, such as medical treatment and autonomous driving, it is risky if we apply few-shot semantic segmentation methods directly.

### 1.2    Visualization of Extensive Ablative Analysis

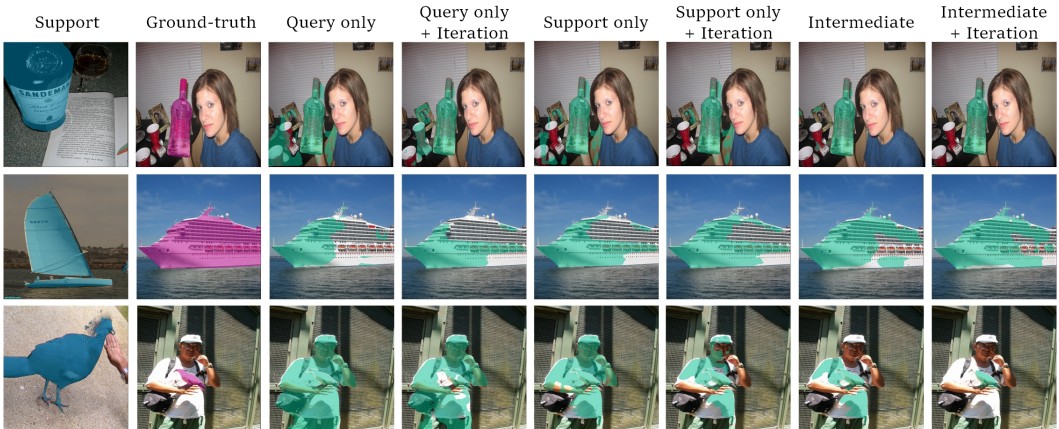

Figure 1: Visualization of extensive ablative analysis on the effectiveness of IPM.

**Effectiveness of IPM.**    As a supplement, we show some qualitative comparison results to prove the effectiveness of our IPM in an intuitionistic way. From left to right in Figure 1, the $3^{rd}$ and $4^{th}$

36th Conference on Neural Information Processing Systems (NeurIPS 2022).

columns show the predictions of only using the query feature to update the learnable prototype. We find that the predicted masks are often confused with other distracting objects. Predictions generated by only using the support feature are shown in the $5^{th}$ and $6^{th}$ columns, which also can not handle the intra-class difference between support and query images. Using the intermediate prototype proposed in IPM can relieve the bias of the unidirectional category knowledge. The last column shows that we get the best results when using the intermediate prototype with iterations.

**Visualization of Using Different Numbers of IPMT Layers.** We further show some qualitative results of stacking different IPMT layers in Figure 2 to prove the effectiveness of our proposed iterative prototype mining scheme. From the $3^{rd}$ column to the $7^{th}$ column, we observe that the accuracy of the predictions is gradually improved by stacking more IPMT layers and the distraction from other objects is relieved. When setting the number of the layers to five, the prediction quality is the best.

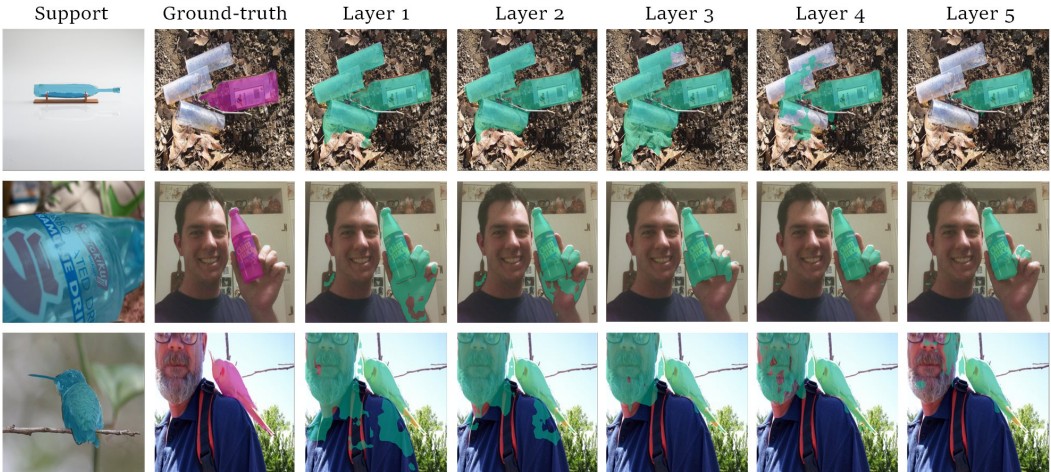

Figure 2: Visualization of the IPMT results by stacking different layers.

## 1.3 Additional Qualitative Results

We give more qualitative results in Figure 3 to show the good performance of our IPMT.

support   query   predict    support   query   predict

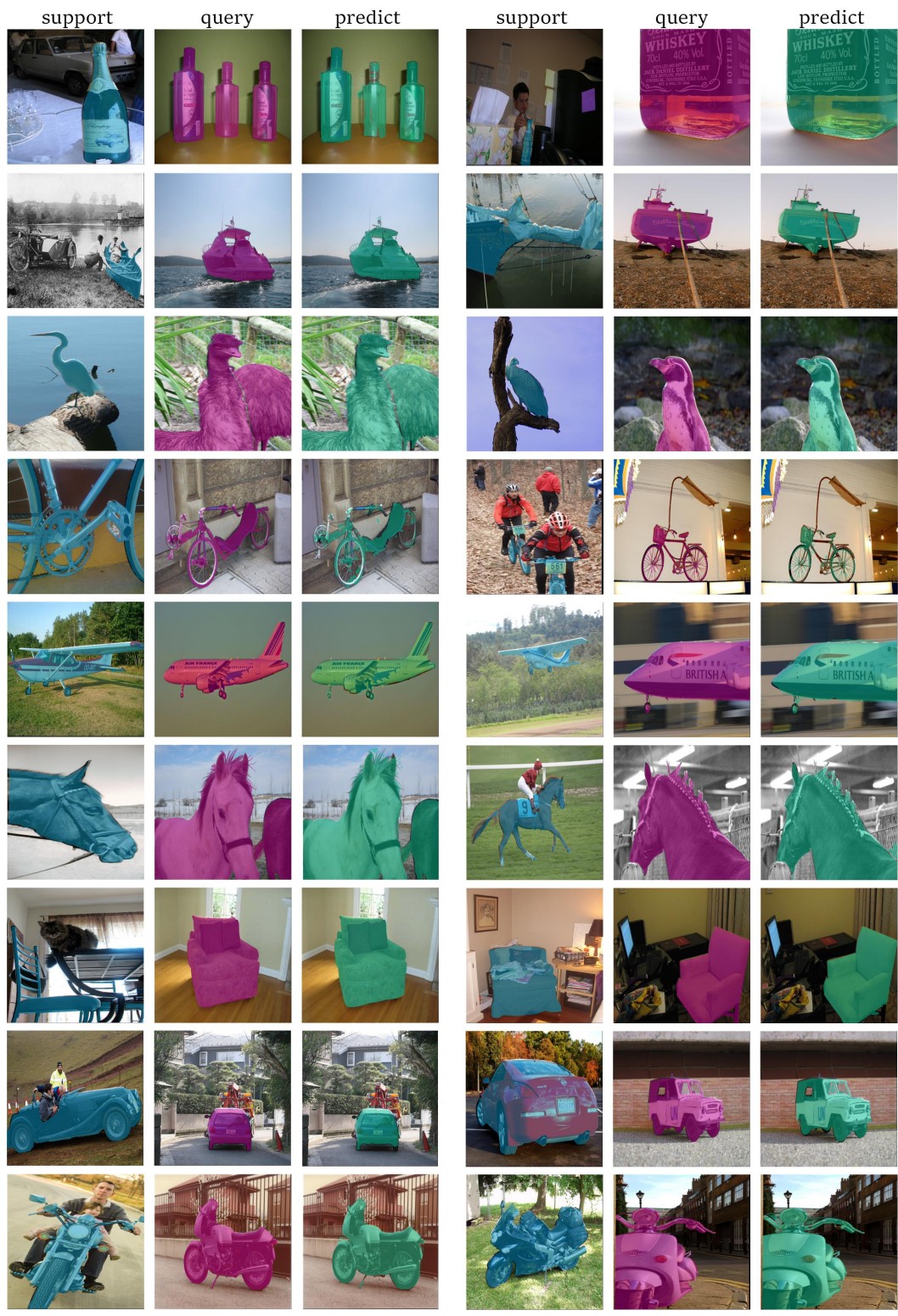

Figure 3: More qualitative results of our IPMT.