# OpenReview forum: "Intermediate Prototype Mining Transformer for Few-Shot Semantic Segmentation"
_NeurIPS.cc/2022/Conference — NeurIPS 2022 Accept_

### Official Review · Reviewer_AjWY · 2022-07-08

**Rating:** 7
**Confidence:** 4
**Soundness:** 3 good
**Presentation:** 3 good
**Contribution:** 3 good

**Summary:**

This paper proposes using the intermediate prototype iteratively generated by Transformer (Intermediate Prototype Mining Transformer, IPMT) to guide the query segmentation. Specifically, it considers to mine category information from both support and query so that the query feature could be better activated. Experiments are performed on the commonly used Pascal-$5^i$ and COCO-$20^i$ datasets.

**Questions:**

See Weaknesses for most questions.

Please address these questions in the rebuttal, especially point 2. I feel confused about why the prototypes would bias toward the query in the symmetrical design. Is it because the prototypes absorb the context information of the query image during the coarse-to-fine procedure?

Besides, some minor points that could be improved:
1. The paper only explains the setting of the 1-shot but does not explain how to extend to K-shot settings.
2. SDL->DSL at the line.296

---
EDIT
The rebuttal resolves my concerns and also provides some interesting analyses. I would like to slightly increase my score, from 6 to 7.

**Limitations:**

Limitations were addressed in the Supplementary.

**Strengths And Weaknesses:**

**Strengths**
1. The motivation of exploring intra-class diversity between support and query makes sense to me.
2. The generating process of the intermediate prototype is well explained, and the iterative design of IPMT is reasonable.
3. According to the experiment part, the proposed method obtains significant improvement compared with the previous method.

**Weaknesses**
1. The baseline in Tab.4 and Tab.5 is not clear. For Tab.4, are DSL and QA applied? For the baseline of Tab.5, it refers to PFENet [28] in line.298, which seems not true according to the description and the result.

2. Since the proposed IPMT processes support and query symmetrically, why would the produced intermediate prototype bias towards the query (as Fig.1 and Fig.4 shown)?

---

> ### Author Response · Authors · 2022-08-02
> **Response to Reviewer AjWY**
>
> Thank you for your constructive review. Here are our responses.
>
> **Q1. The explanation for Tab.4 and Tab.5**
>
> For Table 4, all the ablation studies are conducted with both DSL and QA. In our model, DSL is designed to facilitate the learning of the mined prototype by making sure it can obtain good segmentation performance on both query and support. QA is used to update the query feature. Thus, in Tab.4, for a fair comparison, we use both DSL and QA for all experiments to guide the learning of the prototype and the query feature.
>
> For Table 5, we clarify that the baseline is not the PFENet. As stated in Section 6, our model uses the prior mask and the support prototype to guide the coarse localization. This is the same with PFENet and shown as prototype activation (PA) in Figure 2. However, we did not use PFENet's Feature Enrichment Module in our model. In Line 290, we have stated that we remove IPM, DSL and QA from our model as the baseline. Hence, our baseline only has the PA part and PFENet additionally has the Feature Enrichment Module. We will make this clear in the final version.
>
> **Q2. Bias towards the query**
>
> We argue that our mined intermediate prototype is not biased towards the query. On the contrary, due to the optimization of Eq.(10), it is more biased towards the support since we set $\alpha=0.3$ by considering that the support information is more reliable. Furthermore, to provide quantitative proof, we adopt the Euclidean distance as a metric to measure the distances $D_{qs}$ between the query prototype and the support prototype, $D_{qi}$ between the query prototype and the intermediate prototype, and $D_{is}$ between the intermediate prototype and the support. The average distances of all the categories on the PASCAL dataset are shown below.
>
> | Distance| mean |
> |:------------:|:--------:|
> | $D_{qs}$    | 7.928    |
> | $D_{qi}$    | 6.968    |
> | $D_{is}$    |  4.333    |
>
> From the table, we can clearly see that $D_{qi}$ is smaller than $D_{qs}$ on the whole dataset, which means that our method effectively reduces the distance between the mined prototype and the query. We can also see that $D_{qi}$ is larger than $D_{is}$, which means that our mined intermediate prototype is still biased to the support.
>
> In our paper, Fig.1 and Fig.4 are mainly used to highlight the good performance of our proposed method when facing diverse support that is very dissimilar with the query. To objectively prove this, we measure the corresponding distance on only diverse samples, i.e., $D_{qs}^{div}$, $D_{qi}^{div}$, $D_{is}^{div}$, and report the results below. The diverse samples are defined as those whose $D_{qs}$ is 1.5 times larger than the mean value. The average distances of each category (c1 to c20) and the mean of the whole dataset are shown in the below table.
>
> | Category | c1 | c2 | c3 | c4 | c5| c6 | c7 | c8| c9 | c10 | c11 | c12 | c13 | c14| c15 | c16| c17| c18 | c19 | c20 | mean |
> |:------------:|:------:|:------:|:------:|:------:|:------:|:------:|:------:|:------:|:------:|:-------:|:-------:|:-------:|:-------:|:-------:|:-------:|:-------:|:-------:|:-------:|:-------:|:-------:|:--------:|
> |$D_{qs}^{div}$   | 14.789  | 13.807  | 11.443  | 15.459  | 13.936  | 11.745 | 13.948 | 13.062  | 15.697  | 11.445  | 13.702  | 10.702  | 10.562  | 12.125  | 14.312  | 21.232  | 15.306  | 19.513  | 17.434  | 16.183  | 14.320  |
> | $D_{qi}^{div}$   | 12.008  | 12.215  | 9.651   | 11.371  | 12.198  | 8.277 | 10.149 | **8.369**   | 10.417  | 9.915   | 9.751| 7.956 | 7.419 | 9.762   | **8.642**   | **12.606**  | **10.062**  | **12.683**  | 12.244  | **10.925** | 10.331   |
> |$D_{is}^{div}$    | 6.356   | 6.030   | 5.606   | 10.707  | 6.380   | 7.864   | 8.497   | 9.809   | 10.414  | 5.032   | 7.535   | 6.756   | 6.892   | 6.176   | 9.759   | 14.823  | 11.359  | 12.805  | 11.521  | 11.077  | 8.770    |
>
> We can observe that $D_{qi}^{div}$ is significantly smaller than $D_{qs}^{div}$, which demonstrates that our method achieves remarkable performance on diverse samples. In most categories, $D_{qi}^{div}$ is still larger than $D_{is}^{div}$. However, in six categories (Bold), we found that $D_{qi}^{div}$ is even smaller than $D_{is}^{div}$, which means the intermediate prototype is biased towards the query.
>
> In conclusion, the intermediate prototype is generally more biased to the support. However, for diverse samples, it strives more to approach the query. We are sorry for this misunderstanding and will clarify it.
>
> **Q3. Extension to the K-shot setting**
>
> For the K-shot setting, we use the mean operation on the $\mathbf{MaskAttn}$ results from the k support images. Eq.(6) will be modified as:
> \begin{equation*}
>         \mathbf{IPM}(\mathbf{G},\mathbf{F^s},\mathbf{F^q},\mathbf{M^s},\mathbf{P^q}) =  \mathbf{MLP}(\frac{1}{K} \sum_{j = 1}^{K}  \mathbf{MaskAttn}(\mathbf{G},\mathbf{F^s_j},\mathbf{M^s_j}) + \mathbf{MaskAttn}(\mathbf{G},\mathbf{F^q},\mathbf{P^q})+\mathbf{G}).
> \end{equation*}
> We will make this clear.

---

> > ### Comment · Reviewer_AjWY · 2022-08-06
> > **Reviewer response**
> >
> > I thank the authors for their response.
> > The provided analysis of the intermediate prototypes largely resolves my concern about Fig. 1 and Fig. 5. According to the authors' response, the intermediate prototypes work better than the support prototypes because $D_{qi}$ is smaller than $D_{qs}$, although most intermediate prototypes are still close to the support prototypes. This makes sense to me and would greatly avoid misleading if the author included these explanations in the paper. In addition, I have some questions about this point after reading the rebuttal.
> > * First, the authors define the "diverse samples" as those whose $D_{qs}$ is 1.5 times larger than the mean value in their response. Why chose the value "1.5" and how many samples are categorized as "diverse samples" based on the given criteria? The authors find that the intermediate prototypes are biased towards the query for some categories with "diverse samples". Would the conclusion differ when varying this threshold?
> > * Besides, according to the response, it seems that $\alpha$ in Eq. (10) is essential for learning the intermediate prototypes. Have the authors tried to vary the value? For instance, set $\alpha$ 0.5 or 0.7 to see how $D_{qi}$ and $D_{si}$ change?
> >
> > BTW, just a minor reminder, for Q1, according to Fig. 2 and the code in supplementary, the baseline in Table 5 seems CyCTR without CyCTransformer rather than PFENet without  Feature Enrichment Module since PA is applied to both support and query. Correct me if I am wrong.

---

> > > ### Author Response · Authors · 2022-08-09
> > > **Further Response to Reviewer AjWY**
> > >
> > > Thank you so much for your continued interest and positive responses to our work. Here are further responses to your concerns.
> > >
> > > **Q1. Threshold of diverse samples**
> > >
> > > In the previous rebuttal, we set 1.5 as a threshold to define diverse support and 8.2\% samples are categorized as "diverse support". To further address the reviewer concerns about diverse samples, we provide more average results of all categories about $D_{qs}^{div}$, $D_{qi}^{div}$ and $D_{is}^{div}$ under different thresholds and count the number of categories ($N_q^{div}$) where $D_{qi}^{div}$ is smaller than $D_{is}^{div}$ in the below table. In addition, the mean proportion of diverse samples among all categories is also reported ($Rate$). All the experiments are conducted on PASCAL datasets under 1-shot setting.
> > >
> > > |    Threshold    |   1.1   |   1.2   |   1.3   |   1.4   |   1.5   |
> > > |:---------------:|:-------:|:-------:|:-------:|:-------:|:-------:|
> > > | $D_{qs}^{div}$  | 11.123  | 11.775  | 12.491  | 13.292  | 14.320  |
> > > | $D_{qi}^{div}$  |  9.050  |  9.319  |  9.675  | 10.015  | 10.331  |
> > > | $D_{is}^{div}$  |  6.374  |  6.854  |  7.361  |  7.967  |  8.770  |
> > > |      $Rate$     |  34.01  |  24.85  |  17.775 |  12.085 |   8.2   |
> > > |   $N_q^{div}$   |    0    |    0    |    3    |    3    |    6    |
> > >
> > > From the table, we can see that, with the increment of the threshold from ‘1’ to ‘1.5’, fewer support samples are categorized as "diverse support" according to the $Rate$. However, on the contrary, the number of categories where $D_{qi}^{div}<D_{is}^{div}$ is increasing from 0 to 6 with the threshold increasing. This again proves that our method has significant advantages in dealing with the intra-class diversity problem, especially in diverse support.
> > >
> > > **Q2. Distance for different values of $\alpha$**
> > >
> > > To further illustrate the influence of $\alpha$ for learning of the intermediate prototype, we compare the mean distance $D_{qs}$, $D_{qi}$ and $D_{is}$ of all categories under different $\alpha$ values (i.e. 0.3, 0.5, 0.7). The results are reported in the below table on PASCAL datasets under 1-shot setting.
> > >
> > > | $\alpha$ | 0.3 | 0.5 | 0.7 |
> > > |:---:|:---:|:---:|:---:|
> > > | $D_{qs}$ | 7.928 | 7.164 | 7.582 |
> > > | $D_{qi}$ | 6.968 | 6.502 | 5.262 |
> > > | $D_{is}$ | 4.333 | 5.721 | 6.538 |
> > > | $D_{qi}$ - $D_{is}$ | 2.635 | 0.781 | -1.275 |
> > > | mIoU | 66.8 | 65.3 | 64.2 |
> > >
> > > In table, we can see that $D_{qi}$ is much larger than $D_{is}$ with the best mIoU score of 66.8 under $\alpha=0.3$. The difference between $D_{qi}$ and $D_{is}$ is also larger under $\alpha=0.3$, which means the intermediate prototype is much closer to support rather than that to query. When $\alpha$ increases to $0.5$, $D_{qi}$ is similar to $D_{is}$ and the difference between $D_{qi}$ and $D_{is}$ is narrowed to $0.782$ with 65.3 mIoU under $\alpha=0.5$. This indicates that the intermediate prototype is close to the middle between query and support. Finally, as $\alpha$ increases to $0.7$, $D_{qi}$ is smaller than $D_{is}$ and the mIoU score decreased to 64.2. The difference between $D_{qi}$ and $D_{is}$ becomes a negative value (i.e. -1.275), which means the intermediate prototype is much closer to query rather than that to support. In conclusion, with the increment of $\alpha$, the difference between $D_{qi}$ and $D_{is}$ becomes smaller and smaller, which means that the intermediate prototype would gradually close to query and move away from support. Simultaneously, this movement causes the intermediate prototype fails to obtain more deterministic category information from the support, and results in performance degradation. Thus, we argue that $\alpha$ could influence the biased choice of the intermediate prototype (i.e. biased toward query or support) and the performance of the model.

---

### Official Review · Reviewer_1sTN · 2022-07-12

**Rating:** 5
**Confidence:** 4
**Soundness:** 2 fair
**Presentation:** 3 good
**Contribution:** 2 fair

**Summary:**

This paper deals with the few-shot semantic segmentation problem. Instead of aggregating support category information as prototype, this paper proposes using an intermediate prototype to encode the semantic information from both support and query. By this way, this paper aims to reduce the intra-class discrepancy between support and query. The query features and the learned intermediate prototype are concatenated, followed by 1x1 and 3x3 convolutions, to generate the query mask prediction. The intermediate prototype and the query mask prediction are iteratively refined and improved. Experiments show the effectiveness of proposed method.

**Questions:**

see the weakness part.

**Limitations:**

The authors didn't discuss the limitations and potential negative societal impact of their work.

**Strengths And Weaknesses:**

Pros:
- The idea is reasonable.
- The experiment results are good.
- The paper is easy to follow.

Cons:
- The idea to iteratively refine the query prediction is proposed by previous works, e.g. CANet [1]. And the QA module is similar to some design in IOM of CANet. Thus, those two parts, in my opinion, have quite limited novel contributions.

- The authors adopt separate weights for support and query attention module in IPMT. I would like to see the result when the weights are shared across support and query attention modules.

[1] CANet: Class-Agnostic Segmentation Networks with Iterative Refinement and Attentive Few-Shot Learning, CVPR 2019.

---

> ### Author Response · Authors · 2022-08-01
> **Response to Reviewer 1sTN**
>
> Thank you so much for acknowledging the strength of our method. We have carefully considered your constructive and insightful comments and here are the answers to your concerns.
>
> **Q1. Similar to CANet ?**
>
> **Difference of the iterative refining:**
> The idea of our iterative learning process is to refine the intermediate prototype and the query feature map in each IPMT layer by combining both the deterministic category information from the support and the adaptive category knowledge from the query. However, the idea of CANet is to refine the predicted segmentation mask only using the query feature. Hence, our idea is totally different from CANet and other previous methods.
>
> **Difference between QA and IOM:**
> Our QA module concatenates the expanded intermediate prototype with the query feature map to activate the target regions in it, which is used to refine the query feature. While in IOM, the query feature is concatenated with the previous predicted mask to generate more accurate mask prediction. From the perspective of the used information source, our QA module uses the category information from both query and support in the intermediate prototype, while only query information is utilized in IOM. Hence, they are totally different.
>
> **Q2. Sharing weights across support and query attention modules**
>
> To address your concern, we made some attempts to share weights across support and query attention modules. Surprisingly, the model achieves better performance with 67.3 mIoU score on the PASCAL dataset under the 1-shot setting. Thanks for your advice and we will continue to explore this in the journal extension and future works.

---

### Official Review · Reviewer_65N8 · 2022-07-12

**Rating:** 6
**Confidence:** 4
**Soundness:** 3 good
**Presentation:** 3 good
**Contribution:** 3 good

**Summary:**

The authors proposed an intermediate prototype mining transformer method for few-shot semantic segmentation. According to experimental results on two widely used datasets, the proposed method achieves promising performance compared with other state-of-the-art deep learning models.

**Questions:**

A basic standpoint of the proposed method is intra-class diversity counts for the few-shot segmentation task. Even though sine scatter plot visualizes the diversity in a subjective way, it's more reasonable to introduce some objective metrics or evaluations for diversity.

**Limitations:**

The authors have adequately addressed the limitations and potential negative societal impact of their work.

**Strengths And Weaknesses:**

Strengths:

1. The paper is overall well organized. Authors systematically introduced the related works, limitations, and potential improvement solutions, which are very helpful to direct reviewers/authors to the specific topic of this paper.
2. The proposed method is well described, especially the functions and advantages of each added module, the organized codes further make this study easy to follow.
3. The visualization in the manuscript and supp files are well complementary to the quantitative metrics listed.

Weakness:

Even though it's a very good point that the proposed method performs consistently better than the compared deep learning models on two datasets, I wonder if the added performance is based on larger-scale parameters of the proposed model than the existing ones? So, it would be better to include a discussion/comparison about the number of parameters of both the proposed and compared methods.

---

> ### Author Response · Authors · 2022-08-02
> **Response to Reviewer 65N8**
>
> Thank you so much for acknowledging the strength of our method. We have carefully considered your constructive and insightful comments and here are our responses to your concerns.
>
> **Q1. Comparison of the number of parameters**
>
> **First**, we follow your advice and compare the number of parameters between our method and other state-of-the-art models. When using ResNet50 as the backbone for all models, the results on the PASCAL dataset are shown in the below table. As for our IPMT model, we report all the results using different IPMT layers, i.e., $L=1$ to $L=5$.
>
> | Method  | RPMMS | PFENet | RePRI | HSNet | CWT   | CyCTR | NERTNet | DCP   | IPMT(L=1) | IPMT(L=2) | IPMT(L=3) | IPMT(L=4) | IPMT(L=5) |
> |---------|-------|--------|-------|-------|-------|-------|---------|-------|-----------|-----------|-----------|-----------|-----------|
> | mIoU    | 56.3  | 60.8   | 59.7  | 64.0  | 56.4  | 64.0  | 64.2    | 62.8  | 64.1      | 64.7      | 65.2      | 65.6      | 66.8      |
> | Params. | 19.6M | 34.3M  | -     | 45.2M | 47.3M | 31.9M | 44.5M   | 34.8M | 39.9M     | 46.2M     | 52.4M     | 58.6M     | 64.8M     |
>
> From the table, we can see that our model has more parameters than previous methods when using more than three layers. However, we argue that our model can also achieve better performance than the previous SOTA method NERTNet when using the similar number of parameters, i.e., when using $L=2$, our IPMT has 46.2M parameters and achieves 64.7 mIoU, while NERTNet has 44.5M parameters and achieves 64.2 mIoU.
>
> **Second**, we want to emphasize that our work is the first to focus on the intra-diversity between query and support and propose an intermediate prototype to mitigate this issue. We simply adopted a straightforward iterative process to boost the quality of the intermediate prototype and relieve the category information gap between support and query. As a good starting point, our work can potentially motivate more future works to focus on the intra-diversity problem and how to mine the intermediate information more effectively and efficiently requires further exploration.
>
> **Third**, to further address your concern, we tried to decrease the number of parameters of our IPMT by sharing weights among different layers. Surprisingly, the mIoU score even increases to 68.7 with 5 IPTM layers on the PASCAL dataset under the 1-shot setting, while the number of parameters keeps unchanged (i.e., 39.9M) when increasing the iteration layers. This result shows the great potential of our method and we will continue to explore this in the journal extension and future works.
>
> **Q2. A metric for diversity**
>
> For evaluating the diversity objectively, we adopt the Euclidean distance as a metric to measure the distance between the query prototype and the support prototype ($D_{qs}$) in each episode. To further demonstrate the effectiveness of our method, we also measure the distance between the query prototype and the intermediate prototype ($D_{qi}$) and the distance between the intermediate prototype and the support prototype ($D_{is}$). The average distances of each category (c1 to c20) and the mean of all the categories on the PASCAL dataset are shown in the below table.
>
> | **Category** | **c1** | **c2** | **c3** | **c4** | **c5** | **c6** | **c7** | **c8** | **c9** | **c10** | **c11** | **c12** | **c13** | **c14** | **c15** | **c16** | **c17** | **c18** | **c19** | **c20** | **mean** |
> |:------------:|:------:|:------:|:------:|:------:|:------:|:------:|:------:|:------:|:------:|:-------:|:-------:|:-------:|:-------:|:-------:|:-------:|:-------:|:-------:|:-------:|:-------:|:-------:|:--------:|
> | $D_{qs}$    | 6.967  | 7.326  | 6.980  | 8.402  | 8.445  | 6.903  | 8.599  | 7.391  | 8.937  | 7.091   | 7.842   | 5.983   | 5.668   | 6.828   | 8.052   | 11.011  | 8.283   | 9.705   | 9.409   | 8.740   | 7.928    |
> | $D_{qi}$     | 6.153  | 6.602  | 6.173  | 7.439  | 8.158  | 5.793  | 7.372  | 6.321  | 8.062  | 6.328   | 6.927   | 5.080   | 4.868   | 6.027   | 6.804   | 9.855   | 6.775   | 9.072   | 7.944   | 7.601   | 6.968    |
> | $D_{is}$    | 3.005  | 3.410  | 3.117  | 4.328  | 4.220  | 3.782  | 4.453  | 3.636  | 5.100  | 3.001   | 4.200   | 2.958   | 2.652   | 3.059   | 4.733   | 7.581   | 5.023   | 7.228   | 5.263   | 5.915   | 4.333    |
>
> From the table, we can clearly see that $D_{qi}$ is smaller than $D_{qs}$ on all categories, which means that our mined intermediate prototype is more similar to the query than the support is. This also demonstrates that our method and the proposed intermediate prototypes can effectively mitigate the intra-class diversity problem.

---

### Meta-Review · Area_Chair_UyBd · 2022-08-24

**Recommendation:** Accept
**Confidence:** Certain

**Metareview:**

All reviewers lean to accept this paper and this is a clear acceptance.

**Award:**

No

---

### Decision · Program_Chairs · 2022-09-14

Accept